# KA-1002, a Novel Lysophosphatidic Acid Signaling Antagonist, Alleviates Bovine Tracheal Cell Disruption and Inflammation

**DOI:** 10.3390/ani10020295

**Published:** 2020-02-13

**Authors:** Hee-su Shin, Miok Kim, Kwang Soo Kim, Yong Ki Min, Chang Hoon Lee

**Affiliations:** 1Bio and Drug Discovery Division, Center for Information-Based Drug Research, Research Institute of Chemical Technology (KRICT), Daejeon 305-600, Koreamiok@krict.re.kr (M.K.); ykmin@krict.re.kr (Y.K.M.); 2Department of Biochemistry, College of Natural Sciences, Chungnam National University, Daejeon 34134, Korea; 3School of Medicine, Chungnam National University, Daejeon 301-747, Korea; 4Department of Microbiology, Chonnam National University Medical School, Gwangju 61468, Korea; kwangsu78@hanmail.net

**Keywords:** bovine respiratory disease, lysophosphatidic acid, antagonist, inflammation

## Abstract

**Simple Summary:**

Bovine respiratory diseases are a key factor reducing the productivity of cows in an industrial livestock environment. We have identified a novel lysophosphatidic acid signaling antagonist, KA-1002, which alleviates lysophosphatidic acid-mediated bovine tracheal cell disruptive tissue injury and inflammation. KA-1002 could be considered a novel therapeutic reagent candidate for maintaining respiratory health in reared cattle.

**Abstract:**

The industrial livestock environment can cause stress and weakened immunity in cattle, leading to microbial infections which reduce productivity. As such, there is a need for an effective therapeutic agent that can alleviate uncontrolled destructive respiratory inflammation. We found that lysophosphatidic acid (LPA), a potent endogenous stress-induced inflammatory agent, causes respiratory tissue damage and triggers inflammation in bovine bronchial cells. LPA also inflames pulmonary bovine blood vessel cells to produce inflammatory cytokines. These findings strongly suggest that LPA is a highly important endogenous material exacerbating bovine respiratory diseases. We further identified a novel LPA-signaling antagonist, KA-1002, and showed that it alleviated LPA-mediated bovine tracheal cell disruption and inflammation. Therefore, KA-1002 could potentially serve as a novel therapeutic agent to maintain physiologically healthy and balanced conditions in bovine respiratory tracts.

## 1. Introduction

Bovine respiratory disease (BRD) is the most common and costly illness affecting beef cattle worldwide [1,2,3], and is caused by various environmental factors such as stress due to crowding, ammonia gas, carbon dioxide, and dust in an enclosed barn, as well as by microbial infections [1,2,3,4]. Endogenous factors associated with the immune system are also critical, where weakened immunity leads to susceptibility to pathogenic invasion [5]. Various endogenous immune regulatory molecules, including lysophosphatidic acid (LPA), play roles in respiratory disease in animals [6]. LPA is a powerful endogenous trigger of inflammation in a number of disease-associated conditions [6,7,8]. It is a phospholipid derivative which acts as a potent mitogen through six widely distributed G-protein-coupled receptors (LPARs) with overlapping specificities [8,9]. LPA is synthesized by autotaxin (lysophospholipase D), which removes the choline group from lysophosphatidylcholine or by synthesis of phosphatidic acid [10,11]. Importantly, LPA plays a key role in pulmonary diseases [6], and serves as a therapeutic target for treating refractory idiopathic pulmonary fibrosis [12], which is caused by dust, air pollution, and microbes as well as endogenous factors such as LPA. In addition, pathogenic angiogenesis is critical for inflammatory diseases including respiratory diseases, and LPA is a potent stimulator of this process in multiple diseases [13,14]. All told, LPA can be considered a critical mediator in pulmonary and respiratory diseases of animals, and LPA signaling could be a therapeutic target for animal pulmonary diseases [15,16]. 

In this study, we identified the novel agent KA-1002 as an LPA signaling antagonist using bovine cell-based high-throughput screening (HTS) methods, and found that it had an alleviating effect on LPA-triggered inflammatory cytokine production and loss of cell junction material such as ZO-1 from bovine tracheal cells. Furthermore, we found that KA-1002 alleviated inflammatory cytokine production from blood vessel endothelium. As such, KA-1002 could represent a novel therapeutic reagent for pulmonary diseases in cattle to alleviate LPA-mediated hyper-activated inflammatory tissue disruption in the trachea, as well as LPA-triggered angiogenesis in the respiratory tract. 

## 2. Materials and Methods

### 2.1. Cell-Based High-Throughput Screening (HTS) 

To identify potent LPA signaling antagonists, we screened compounds from a 2000-compound library at the Korea Research Institute of Chemical Technology using an HTS system based on cellular analysis of the LPA-induced inflammatory cytokine TNFα production using the CAPE bovine blood vessel cell line (ATCC® CCL-209™), from ATCC (Manassas, VA, USA). CAPE pulmonary artery endothelium cells were cultured in complete Roswell Park Memorial Institute (RPMI) 1640 medium supplemented with 10% fetal bovine serum and 1% penicillin streptavidin in a CO_2_ incubator at 95% air, 5% CO_2_, and 37 °C. For analysis of TNFα expression, CAPE cells were treated with 100 µM LPA (Sigma-Aldrich, St. Louis, MO, USA) in the presence of each compound at 20 µM, from the compound library described above. Total RNA was extracted using TRIzol reagent (Invitrogen, Carlsbad, CA, USA), and 20 ng was used as the template for real-time reverse transcription PCR along with Thermoscript^TM^ RT-PCR cDNA SuperMix (Invitrogen, Carlsbad, CA, USA). Following this, Brilliant II CYBR® Green qPCR Master Mix was used for PCR (Agilent Technologies Inc., Santa Clara, CA, USA). Detailed qPCR procedure is described as below. 

### 2.2. Drugs and Chemicals

KA-1002 (Figure 1A) was dissolved in 100% DMSO at a concentration of 30 mg/mL as a stock solution stored at −20 °C and diluted in complete culture medium prior to experiments. LPA was purchased from Sigma-Aldrich Co Ltd. Ki-16425, which is a known LPA-antagonist, was also purchased from Sigma-Aldrich Co Ltd.

### 2.3. Cell Culture

EBTr (NBL-4; ATCC® CCL-44™) bovine tracheal cells were purchased from ATCC and cultured in complete Roswell Park Memorial Institute (RPMI) 1640 medium supplemented with 10% fetal bovine serum and 1% penicillin streptavidin in a CO_2_ incubator at 95% air and 5% CO_2_ at 37 °C. CAPE (ATCC® CCL-209™) pulmonary artery endothelial cells were purchased from ATCC and cultured in complete RPMI 1640 medium under the same conditions. For analysis of ZO-1 expression, EBTr cells were treated with 100 µM LPA (Sigma-Aldrich) in the presence or absence of 20 µM KA-1002 in triplicate for 18 h in the media and conditions described above.

### 2.4. Tube Formation Assay 

For the tube formation assay, CAPE cells (ATCC® CCL-209™) from ATCC were seeded at a density of 3.5 × 10^5^ cells/mL onto growth factor reduced Matrigel (Corning, New York, USA) in an ibidi μ-plate Angiogenesis 96 Well (Ibidi, Gräfelfing, Germany). CAPE were treated with 100 μM LPA purchased from Sigma-Aldrich Co Ltd with or without 20 μM KA-1002 in RPMI 1640 medium supplemented with 10% fetal bovine serum and 1% penicillin streptavidin in a CO_2_ incubator at 95% air and 5% CO_2_ at 37 °C. Cells were incubated for 3 days. Light field images were obtained with an inverted microscope (IX51by Olympus, Tokyo, Japan). The tube characteristics were analyzed and quantified using Imaris software (Oxford Instruments, Abingdon-on-Thames, UK).

### 2.5. Cell Adhesion Analysis and Survival Rate (FACS Analysis) 

Briefly, 3 × 10^3^ EBTr cells were seeded in 96-well cell culture plates (SPL Life Sciences, Pocheon, Korea). After 24 h, cells were treated with 100 µM LPA (Sigma-Aldrich) in the presence or absence of 5 or 20 µM KA-1002 in triplicate. For analysis of cell adhesion, adherent and unattached cells were counted using an IX51 microscope (Olympus Optical Co., Center Valley, PA, USA). Images were processed and anayyzed using Imaris software (Oxford Instruments). For analysis of the survival rate of EBTr cells, cells were harvested and stained with 7-aminoactinomycin D (7-ADD) ((Thermo Fisher Scientific, Waltham, MA, USA). 7-ADD positive cells (dead cells) and 7-ADD negative cells (live cells) were analyzed using FACS analysis. Staining data were collected using a MACSQuant VYB (Miltenyi Biotec, Bergisch Gladbach, North Rhine-Westphalia, Germany).

### 2.6. Total RNA Isolation and Real-Time Reverse Transcription PCR

EBTr and CAPE cells were prepared as described above. Total RNA was isolated from cells and cDNA was generated as previously described. Primer and probe sets (Table 1) were purchased from Bioneer Corporation (Daejeon, Korea) Results were normalized to the level of β-actin. Real-time qPCR was performed with duplicate samples using an Agilent Technologies AriaMx real-time system (Agilent Technologies Inc.). For cells from each donor, relative expressions based on 2^−ΔΔCT^ values are shown as percentages relative to values obtained for the subset with the highest expression.

### 2.7. Immunocytochemistry

EBTr cells were seeded on an 8-well μ-slide (Ibidi, Planegg, Germany) and cultured for 24 h in the absence or presence of LPA or KA-1002 at 37 °C and 5% CO_2_, then fixed with 3.7% paraformaldehyde for 20 min. The slides were washed and permeabilized with PBS containing 0.01% Triton X-100 for 30 min, followed by blocking with PBS containing 1% bovine serum albumin for 1 h at room temperature. The cells were then incubated overnight at 4 °C with fluorophore-conjugated primary antibodies. The following day, cells were washed three times with 1% bovine serum albumin for 10 min and stained with DAPI diluted 1:1000 in PBS for 10 min. Primary antibodies against ZO-1 (Thermo Fisher Scientific) and phalloidin (Thermo Fisher Scientific) for F-actin were both diluted 1:200 and used to visualize expression of ZO-1, actin filament-based intercellular structures, and tight junctions. Fluorescent signals were imaged with an LSM 880 confocal laser scanning microscope equipped with VIS and NIR lasers. All captured images were taken using the Airyscan mode supported by the LSM 880 confocal laser scanning microscopy for image optimization (Carl Zeiss, Oberkochen, Germany).

### 2.8. SwissADME Analysis

Physiochemical properties of KA-1002 were computed and predicted for ADME parameters, pharmacokinetics properties, druglike nature, and medicinal chemistry friendliness using Swiss ADME analysis (Swill Institute of Bioinformatics).

### 2.9. Statistical Analysis

Multiple comparisons were carried out using ANOVA. Differences between two groups were evaluated using an unpaired t-test. Statistical analyses were performed using Prism software (GraphPad Inc., San Diego, CA, USA), with differences considered significant at *p* < 0.05.

## 3. Results

### 3.1. The LPA-Antagonistic Novel Compound KA-10002 Works against LPA-Induced Inflammatory Cytokine Production from Bovine Pulmonary Blood Endothelial Cells 

In this study, we screened inhibitors of cellular analysis based on LPA-induced inflammatory cytokine, TNFα production using the bovine blood vessel cell line CAPE from a 2000-compound library at the Korea Research Institute of Chemical Technology, which were chosen based on their structural homology with a well-known LPA antagonist, Ki-16427, using an HTS system. Of all compounds screened, we identified 14 with inhibitory effects against LPA greater than 70% at 5.0 µM; compound KA-1002, shown in Figure 1A, was the most potent. Its chemical structure was identified as (2R,3R,4S,5R)-2-(6-((2-hydroxyethyl)amino)-9H-purine-9-yl)-5-(hydroxymethyl)-tetrahydrofuran-3,4-dio) (Figure 1A). We then examined the effects of KA-1002 on other inflammatory cytokines and chemokine production from LPA-inflamed CAPE cells. As shown in Figure 1B–D, KA-1002 was shown to decrease the induction of TNFα, IL-6, and IL-1β expression in LPA-treated CAPE cells in a dose-dependent manner. Furthermore, we predicted physiochemical properties, ADME parameters, pharmacokinetic properties and drug likeness of KA-1002 using the Swiss ADME program. As a result, we found that KA-1002 compound showed drug likeness without any violation of ADME parameters based on Lipinski’s rule of five (Figure 1E). Lastly, we compared the relative expression level of LPA receptor, LPAR1, and LPAR2 on CAPE cells (Figure 1F) showing higher expression level of LPAR1 rather than LPAR2. This compound was first reported as an LPA antagonist in this study. To compare the LPA-antagonistic effect of KA-1002, we used Ki-16425 which was known as an LPA antagonist. In this study, we found that our novel compound, KA-1002, showed a similar level of LPA antagonistic effect as Ki-16425 (Figure 1B–D). 

### 3.2. KA-10002 Alleviates LPA-Induced Inflammatory Angiogenesis

As previously reported [13], it was well-known that LPA triggers angiogenesis in inflammatory sites to amply inflammation. Because KA-1002 inhibited inflammation in bovine blood vessel cells, we hypothesized that it might affect LPA-mediated increased angiogenesis. LPA treatment increased vessel formation in CAPE cells compared to untreated controls (Figure 2A), but this effect was significantly reduced by KA-1002 treatment in a dose-dependent manner (Figure 2B). This indicates that KA-1002 can alleviate inflammatory cytokine production and increased angiogenesis induced by LPA in inflamed bovine blood vessel cells.

### 3.3. KA-10002 Alleviated LPA-Induced Downregulated Expression and Disruption of Cellular Arrangement Of ZO-1 in EBTr Cells 

LPA strongly reduced the expression of ZO-1 protein in EBTr cells. Untreated healthy cells highly expressed ZO-1 protein in the whole cell and cellular boundary region, while cells treated with 100 µM LPA showed significantly reduced ZO-1 expression. KA-1002 ameliorated this outcome (Figure 3A). Relative amounts of ZO-1 protein in untreated, LPA-treated, and LPA + KA-1002-treated EBTr cells were analyzed using ImageJ imaging analysis software to show the alleviating effect of KA-1002 on LPA-induced loss of ZO-1 in EBTr cells (Figure 3B). F-actin expression was not significantly changed by LPA or KA-1002 (Figure 3C). We measured the relative amount of ZO-1 and F-actin per each cell in the same fields of the image. Specifically, LPA treatment disrupted cellular arrangement of ZO-1 protein in EBTr cells (Figure 3A). In untreated EBTr cells, ZO-1 protein was located in the whole cytosol and the marginal area of cells. Interestingly, ZO-1 expression is higher in the cellular boundary region. However, LPA-treated EBTr cells lost ZO-1 protein in most of the cellular boundary region (Figure 3A). Importantly, KA-1002 treatment on LPA-treated EBTr cells recovered ZO-1 protein arrangement in the cellular boundary region. These results strongly suggest that LPA treatment might induce loss of cell-to-cell tight junctions and cell-to-matrix adhesion, because boundary arranged ZO-1 protein is important for maintaining cell-to-cell tight junctions and cell-to-matrix adhesion.

### 3.4. KA-10002 Alleviated LPA-Induced Loss of Adhesive Capacity in Bovine Tracheal Cells

LPA-treated EBTr cells showed significantly less adhesion to culture plates compared to controls (Figure 4A), and the ratio of unattached cells compared to adherent cells was greater in LPA-treated cells (Figure 4B). Meanwhile, KA-1002 led to a significant recovery in the ratio of unattached versus adherent cells (Figure 4B). However, the percentage of dead cells was not significantly different among untreated, LPA-treated, or KA-1002 + LPA-treated cells (Figure 4C). These results suggest that LPA reduced cell-to-cell tight junctions and cell-to-matrix adhesion, which could cause structural disruption or tissue damage. KA-1002 support of both ZO-1 expression and cell attachment indicates that it may be beneficial for cell-to-cell tight junction and cell-to-matrix adhesion of bovine bronchial cells. 

### 3.5. KA-10002 Alleviated LPA-Induced Inflammatory Cytokine Production 

To characterize LPA-mediated inflammatory cytokine production in bovine bronchial cells, we measured TNFα, and IL-1β from untreated, LPA-treated, and KA-1002 + LPA-treated EBTr cells. TNFα and IL-1β were significantly increased by LPA treatment, but these effects were strongly inhibited by KA-1002 (Figure 5A,B). Given the similar findings in endothelial CAPE cell culture, this suggests that LPA triggers inflammation in bovine bronchial and endothelial cells by production of inflammatory cytokines, while KA-1002 worked to counteract all of these outcomes.

## 4. Discussion

LPA regulates a broad range of physiological and cellular responses and has various synthesis pathways [6,8,13]. Among these, ATX is an enzyme responsible for the bulk of LPA production in the plasma and at inflamed and/or malignant sites [17]. Lysophosphatidylcholine (LPC), the main substrate of autotaxin, is a highly abundant bioactive lysoglycerophospholipid present in circulation at high concentrations (100-200 μM) [15,17,18]. Multiple LPA effector functions are at least partially mediated by six G-protein-coupled LPA receptors (LPARs) with overlapping specificities and widespread distribution. In addition, a group of transmembrane lipid-phosphate phosphatases have been demonstrated as negative regulators of LPA metabolism [8]. Beyond the well-established role of the ATX/LPA axis in carcinogenesis, high levels of autotaxin expression have frequently been observed in non-malignant, inflamed tissues, suggesting a reasonable involvement of ATX in chronic/acute inflammatory disorders [9,10,19]. Many ATX/LPA antagonists and LPAR inhibitors have been developed for therapeutic purposes. As such, the ATX/LPA axis is one of the most important therapeutic targets in humans for multiple inflammatory diseases, including respiratory illnesses [15]. Our results suggest comparable therapeutic targets for respiratory diseases in livestock cattle [6,10]. 

LPA is highly abundant in multiple respiratory diseases and blocking ATX/LPA signaling could alleviate those diseases [6,19]. Even though the pathological roles of LPA in cows have not been fully investigated in vivo until now, many in vitro studies in other animal models, including humans, strongly suggest that LPA importantly works in multiple respiratory diseases [6,12,19]. Furthermore, there are interesting in vivo studies in cattle for development of reproductive diseases and development of embryos [20,21]. These reports strongly suggest that LPA might have a critical role in cows and should be investigated in comparison with other well-investigated models, such as mouse and human. In industrial animal production, respiratory diseases are a major threat caused by multiple factors including pathogens and environmental conditions. The most common microbes associated with BRD are not highly aggressive or severe pathogens in healthy cows, and are common in the environment. However, stress due to the conditions of an enclosed barn, including atmospheric ammonia, CO_2_, dust, temperature fluctuations, and crowding can lead to immune depression and susceptibility to these pathogens, reducing productivity [1]. There is a need for an effective therapeutic agent that can enhance resistance against common respiratory diseases in cattle.

Downstream of LPA receptor activation, the small GTPase Rho can be activated, subsequently activating Rho kinase [22,23] and leading to the formation of stress fibers and cell migration via inhibition of myosin light-chain phosphatase. These results strongly suggest that LPA signaling can affect cell migration, cell adhesion, and cell-to-cell junction formation [14,16,24,25]. We found that LPA, a critical endogenous inflammatory material causing tissue damage in respiratory disease, disrupted the expression of ZO-1 protein in bovine bronchial cells, which should make host animals highly susceptible to common pathogenic microbes and respiratory inflammation (Figure 3). We also observed that LPA-treated bovine bronchial cells lost cell adhesion (Figure 4A), and that the ratio of unattached to attached cells was greatly increased in LPA-treated bovine bronchial cells (Figure 4B). This might be explained by LPA receptor activation of Rho kinase and Rho-mediated cell transformation into epithelial phenotypes and mesenchymal phenotypes [25]. More importantly, as an LPA signaling antagonist, KA-1002 treatment led to significantly recovered cell adhesion (Figure 4B) and expression of ZO-1 (Figure 3B). These findings suggest that KA-1002 alleviates LPA signaling activation-mediated disruptive responses in bovine bronchial cells. 

We also found that LPA strongly induced the inflammatory cytokines TNFα, IL-6, and IL-1b, causing pathological inflammation in bovine bronchial cells and endothelial cells with pathological angiogenesis, suggesting that a high degree of LPA activity is detrimental to respiratory tract health. Meanwhile, KA-1002 clearly showed potential as a therapeutic agent for alleviating LPA-mediated bovine tracheal cell disruption and inflammation. In this study, we also compared KA-1002 with a known LPA antagonist, Ki-16425, which had some structural homology to KA-1002. However, our compound KA-1002 showed, in Figure 1, a similar level of Ki-16427 in the LPA antagonistic effect. The LPA antagonistic effect of KA-1002 might be caused by structural similarity with Ki-16427, but KA-1002 is a novel compound as an LPA antagonist. As such, ATX, LPA, and LPA receptors might represent therapeutic targets for respiratory diseases of cows, and KA-1002 could be a potential therapeutic agent as a drug or feed additive against multiple respiratory diseases. It could also be used in combination with antibiotics or antiviral reagents to treat pathogen-mediated or commensal microbe-mediated respiratory diseases. We did not demonstrate the therapeutic effects of KA-1002 in vivo, which would be a logical next step in confirming its applicability. Nonetheless, our results give a strong indication that KA-1002 represents a novel class of potential therapeutic agents for treating bovine respiratory disease. This study is the first study to suggest that LPA antagonists could be a target for treating respiratory diseases of industrial cows, as well as the first report of KA-1002 as a novel LPA antagonist that alleviates LPA-mediated bovine tracheal cell disruption.

## 5. Conclusions

According to the results, it could be concluded that KA-1002 as a novel LPA antagonist alleviates LPA-mediated bovine tracheal cell disease associated cellular responses, such as inflammatory angiogenesis and inflammatory cytokine production. Further studies using cows are needed to confirm the therapeutic efficacy of KA-1002, whose in vivo efficacy was established. 

## Figures and Tables

**Figure 1 animals-10-00295-f001:**
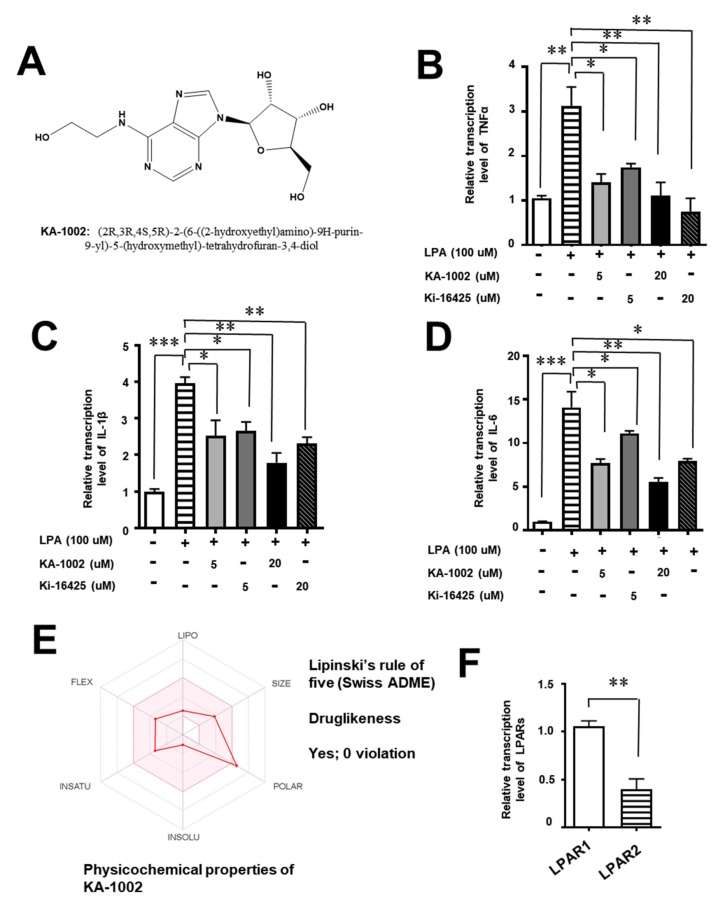
(**A**) Chemical structure of KA-1002, an inhibitor screened from cellular analysis based on lysophosphatidic acid (LPA) induced inflammatory cytokine, TNFα production using bovine blood vessel cell line, CAPE. Inhibitory effects of KA-1002 on the production of LPA-induced (**B**) TNFα, (**C**) IL-1β, and (**D**) IL-6 production in bovine endothelial cells, CAPE. The cells were exposed to KA-1002 and Ki-16425 at 5 and 20 μM in the presence of 100 μM of LPA for 18 h. Inhibition rates were calculated by comparing the corresponding levels of TNFα from the control (LPA-untreated) group under the high-throughput (HTS) system. Pooled results are shown from 3 independent experiments. ns: not significant; **p* < 0.05, or ***p* < 0.01, indicate significant differences in comparison to the LPA-treated group according to two-tailed unpaired Student’s t-testing. Error bars denote standard error (SEM). (**E**) Physiochemical properties of KA-1002 based on Swiss ADME analysis. (**F**) Relative transcriptional level of LPAR1 and LPAR2 on CAPE cells. Pooled results are shown from 3 independent experiments. ** *p* < 0.01.

**Figure 2 animals-10-00295-f002:**
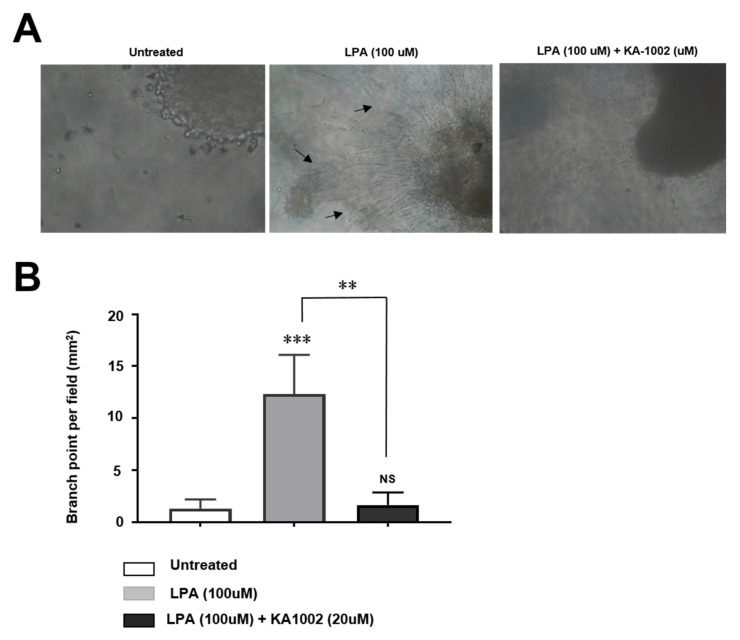
Inhibitory effects of KA-1002 on LPA-induced angiogenesis of bovine blood vessel cell line, CAPE. (**A**) Representative images of the CAPE cells exposed to 100 μM of LPA in the presence or absence of 20 μM KA-1002 for 3 days were from three different independent experiments. (**B**) Quantification of tube network was calculated using Imaris software (Oxford Instruments). Pooled results are shown from 3 independent experiments. **p* < 0.05, ***p* < 0.01, or ***p* < 0.001 indicate significant differences in comparison to the LPA-treated group according to two-tailed unpaired Student’s t-testing. Error bars denote standard error (SEM).

**Figure 3 animals-10-00295-f003:**
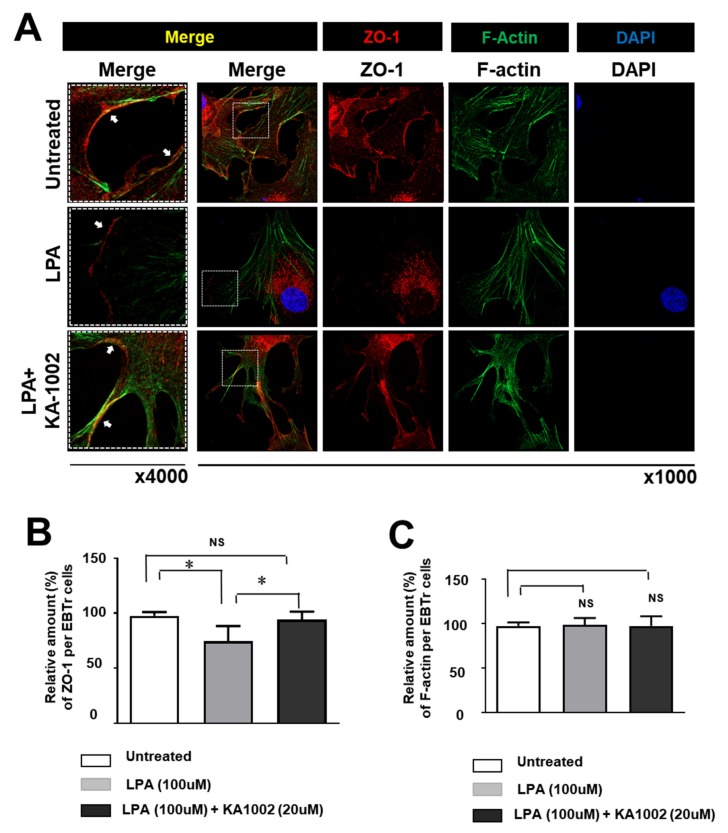
(**A**) EBTr cells were treated with 100 μM LPA or 100 μM LPA + 20 μM KA-1002 for 18 hours. Representative images of ZO-1 expression in each group were obtained using confocal microscopy, with ZO-1 shown in red, F-actin stained with phalloidin in green, and nuclei stained with DAPI in blue (scale bar 50 μm). Representative images from the four different experiments are shown in Figure 3A. Pooled results from four different experiments are shown for relative amounts of (**B**) ZO-1 and (**C**) F-actin. The amount of ZO-1 and F-actin per cell were calculated by dividing the total amount of ZO-1 and F-actin with number of nuclei in the four different images per each treatment group using the ImageJ software. ns: not significant; ** *p* < 0.01 indicates significant differences in comparison to untreated control cells according to two-tailed unpaired Student’s t-testing. Error bars denote standard error.

**Figure 4 animals-10-00295-f004:**
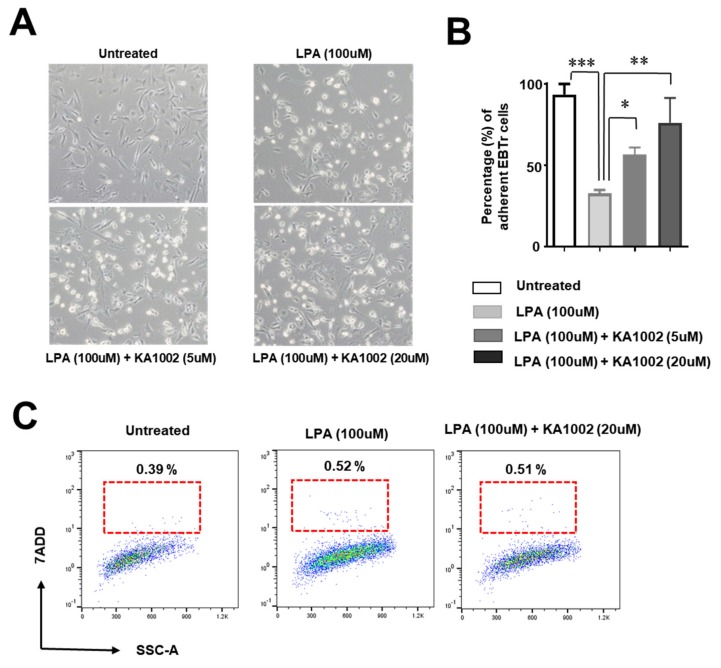
EBTr cells were treated with 100 μM LPA, or 100 μM LPA + 20 μM KA-1002 for 18 h. (**A**) Representative images of each group were obtained using light field microscopy. (**B**) Graphical analysis of the ratio of unattached cells to adherent cells in four different experimental images. * *p* < 0.05, ** *p* < 0.01, or *** *p* < 0.001 01 indicates significant differences in comparison to untreated control cells according to two-tailed unpaired Student’s t-testing. Error bars denote standard error. (**C**) Representative images of FACS analysis of EBTr cells treated with 100 μM LPA, or 100 μM LPA + 20 μM KA-1002 for 18 hours. 7-ADD positive cells were analyzed. These results were representative from three different experiments.

**Figure 5 animals-10-00295-f005:**
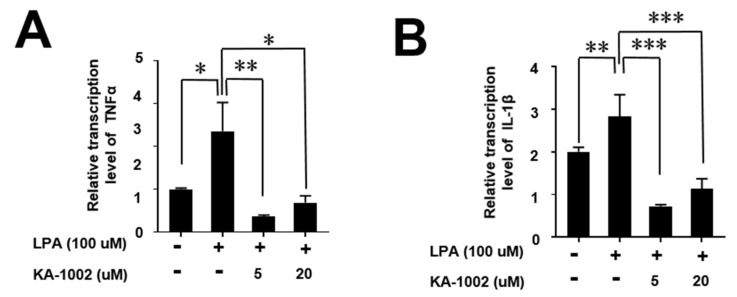
Inhibitory effects of KA-1002 on the production of LPA-induced (A) TNFα and (B) IL-1β production from EBTr cells. The cells were exposed to KA-1002 at 5 and 20 μM in the presence of 100 μM of LPA for 18 hours and inhibition rates were calculated by comparing the corresponding production of respective cytokines from the LPA-untreated control. Pooled results are shown from three independent experiments. * *p* < 0.05, ** *p* < 0.01, or *** *p* < 0.001 indicate significances compared to the LPA-treated group according to two-tailed unpaired Student’s t-testing. Error bars denote standard error.

**Table 1 animals-10-00295-t001:** Primer sequences.

Gene Name	5′-Primer Sequence	3′-Primer Sequence
Bovine TNFα	TCTCTCTCACATACCCTGCCA	CCACATCCCGGATCATGCTT
Bovine IL-6	CCAGCCACAAACACTGACCT	CCCCAGCTACTTCATCCGAA
Bovine IL-1β	AACGTCCTCCGACGAGTTTC	CCAGCACCAGGGATTTTTGC
Bovine LPAR1	AACACAGGGCCCAATACTCG	CAATTGCAATGGCCAGGAGG
Bovine LPAR2	CCACGAGTCTGTTCGCTACA	GTGGCATTTGCTGTACCCTG
Bovine β-actin	TCGGTTGGATCGAGCATTCC	GTGGCTTTTGGGAAGGCAAA

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
