# Peer review of "KA-1002, a Novel Lysophosphatidic Acid Signaling Antagonist, Alleviates Bovine Tracheal Cell Disruption and Inflammation"

_animals, 2020, doi:10.3390/ani10020295_

Round 1

Reviewer 1 Report

I suggest acceptance of the revised manuscript which is significantly improved and conclusions are supported by new experimental work.

Reviewer 2 Report

Questions have been addressed.

This manuscript is a resubmission of an earlier submission. The following is a list of the peer review reports and author responses from that submission.

Round 1

Reviewer 1 Report

Hee-su Shin et al., KA-1002

In this manuscript the authors describe a novel compound that inhibits LPA-mediated signaling events in a bovine tracheal cell line (EBTr) and a bovine pulmonary artery endothelial cell line (CAPE). Given the involvement of LPA-mediated signaling pathways in pulmonary pathologies the results are potentially important. Revision of introduction and focusing of discussion to place the observations in framework of LPAR signaling and therapeutic potential is also important. The following points need attention:

- If available please provide clear evidence that LPA-related pathologies are a real problem for cattle reared in livestock - the evidence provided in the current version covers either mouse or in vitro work.

- Although the authors provide evidence that the compound inhibits several aspects of LPA signaling the mode of action of this inhibitor is completely unclear. Is this an LPAR inhibitor (or PPARgamma at the LPA concentrations used)? Is there structural homology to other LPAR antagonists? Please provide relevant information or discuss.

Nomenclature should be: …-tetrahydrofuran-3,4-diol (Fig. 1)

- Please provide calculated physicochemical properties (e.g. by using SwissADME or another algorithm) that are relevant to Lipinski’s ‘rule of five’

- Which LPA receptors do the two cell lines express? This should be shown and included in the discussion.

- Cytotoxicity measurements must be shown for the inhibitor in a concentration- and time-dependent manner (the WST-1 assay is mentioned in the M&M section but no results are presented).

- ZO-1 immunofluorescence (Fig. 3) is not convincing since there is no evidence for tight monolayer or tight junction formation. Looks more like clusters of loosely adherent cells.

- Are the floating cells shown in Fig. 4 still alive? Please provide evidence, e.g. by flow cytometry or trypan blue exclusion.

Author Response

In this manuscript the authors describe a novel compound that inhibits LPA-mediated signaling events in a bovine tracheal cell line (EBTr) and a bovine pulmonary artery endothelial cell line (CAPE). Given the involvement of LPA-mediated signaling pathways in pulmonary pathologies the results are potentially important. Revision of introduction and focusing of discussion to place the observations in framework of LPAR signaling and therapeutic potential is also important. The following points need attention:

Q1: If available please provide clear evidence that LPA-related pathologies are a real problem for cattle reared in livestock - the evidence provided in the current version covers either mouse or in vitro work.

 Answer: It is a good point, but there are a limited number of studies to show those evidence. Specifically those studies focused on reporductive and embryonic development and disease. As many previous studies in humans and mouse models strongy suggest that pathological roles of LPA in multiple diseases including pulmonary diseases, we should investigate pathological roles of LPA in pulmonary diseases in vivo cattles. We added new references (20, 21) and revised manuscript in line 272-277 on page 10.

Q2: Although the authors provide evidence that the compound inhibits several aspects of LPA signaling the mode of action of this inhibitor is completely unclear. Is this an LPAR inhibitor (or PPARgamma at the LPA concentrations used)? Is there structural homology to other LPAR antagonists? Please provide relevant information or discuss.

Answer: In our revised manuscript, we used a known LPA antagonist, Ki-16427 to compare KA-1002 in several experiment. In fiugre 1, we found that KA-1002 and Ki-16427 showed similar LPA antagnoistic effect. For structural homology issues, KA-1002 has some of structural similarity with Ki-16427. That might be a cause for LPA antagonistic effect of KA-1002. We commented that point in line 304-308 on page 10. And we showed experimental result with Ki-16427 in Figure 1.

Q3: Nomenclature should be: …-tetrahydrofuran-3,4-diol (Fig. 1)

Answer: We revised the nomenculature as you recommented in figure 1.

Q4: Please provide calculated physicochemical properties (e.g. by using SwissADME or another algorithm) that are relevant to Lipinski’s ‘rule of five’

 Answer: We added SwissADME analysis result in Figure 1E. And commented that point in result section in lin 155-158 on page 4.

Q5: Which LPA receptors do the two cell lines express? This should be shown and included in the discussion.

 Answer: We measured LPAR1 and LPAR2 using qPCR, because there are no reliable antibody for LPARs. In our result, we found that LPAR1 is highter transcription than LPAR2. We showed in Figure 1F and mentioned.

Q6: Cytotoxicity measurements must be shown for the inhibitor in a concentration- and time-dependent manner (the WST-1 assay is mentioned in the M&M section but no results are presented).

 Answer: We perfromed FACS analysis (7-ADD staining for live & dead staining).

We showed the results in figure 4C. As we showed, the percentages of dead cells are not changed by LPA treatment and KA-1002. Also, WST-1 analysis is redundant result, so we deleted the WST-1 method in M&M section. Instead, we showed FACS analysis results in Figure 4C and M&M section.

Q7: ZO-1 immunofluorescence (Fig. 3) is not convincing since there is no evidence for tight monolayer or tight junction formation. Looks more like clusters of loosely adherent cells.

 Answer: We replaced former version of Figure 3 with improved images for ZO-1 protein arrangement & actin-filaments. Also we used higher magnitude image (4000X) to show ZO-1 protein arrangements in cellular margin.

Q8: Are the floating cells shown in Fig. 4 still alive? Please provide evidence, e.g. by flow cytometry or trypan blue exclusion.

Answer: We showed the results in figure 4C. As we showed, the percentages of dead cells are not changed by LPA treatment and KA-1002. Also, WST-1 analysis is redundant result, so we deleted the WST-1 method in M&M section. Instead, we showed FACS analysis results in Figure 4C and M&M section.

Reviewer 2 Report

The manuscript by Shin et. al identified KA-1002 as one novel LPA antagonist. It is an interesting study.

The intention of this study was to identify LPA antagonists for potential drug candidates for bovine respiratory diseases. Why did the authors perform the screening using a bovine blood vessel cell line, instead of tracheal cells directly? Could the authors have a brief description of the 2000-compound library at the Korea Research Institute of Chemical Technology? To all fairness, the library is not that large. Why did the authors choose the library to screen for LPA signaling antagonists? Could the compound KA-1002 be available commercially or from the authors for application in other researchers’ studies? Did the authors perform serum starvation of the cells before LPA stimulation? Line 102, “3 × 103 EBTr cells” should be “3 × 103 EBTr cells”? Line 150-151, it is not appropriate to state “IL-6, and IL-1β expression in LPA-treated CAPE cells were significantly reduced by KA-1002 in a 150 dose-dependent manner” , KA-1002 was shown to decrease the induction of  IL-6 and IL-1β expression by LPA only at 20 uM, but not at 5 uM. Fig.5, KA-1002 at lower concentration has stronger inhibition on the induction of TNFa and IL-1b by LPA stimulation in EBTr cells? This is an interesting observation. It would be nice for the authors to compare KA-1002 with other LPA antagonists in the discussion section.

Author Response

First, Thanks for your kind review; We carefully revised our manuscript as below;

Q1: The intention of this study was to identify LPA antagonists for potential drug candidates for bovine respiratory diseases. Why did the authors perform the screening using a bovine blood vessel cell line, instead of tracheal cells directly? Could the authors have a brief description of the 2000-compound library at the Korea Research Institute of Chemical Technology? To all fairness, the library is not that large. Why did the authors choose the library to screen for LPA signaling antagonists? Could the compound KA-1002 be available commercially or from the authors for application in other researchers’ studies? Did the authors perform serum starvation of the cells before LPA stimulation? Line 102, “3 × 103 EBTr cells” should be “3 × 103 EBTr cells”? Line 150-151, it is not appropriate to state “IL-6, and IL-1β expression in LPA-treated CAPE cells were significantly reduced by KA-1002 in a 150 dose-dependent manner” , KA-1002 was shown to decrease the induction of  IL-6 and IL-1β expression by LPA only at 20 uM, but not at 5 uM. Fig.5, KA-1002 at lower concentration has stronger inhibition on the induction of TNFa and IL-1b by LPA stimulation in EBTr cells? This is an interesting observation. It would be nice for the authors to compare KA-1002 with other LPA antagonists in the discussion section. 

Answer: LPA is a well known for regulator in blood vessel endothelial cells. Thus first we hypothesized that LPA should potently regulate inflammatory cytokines from blood endothelial cells. That is the reason which we chose CAPE cells for screening. Korea Research Institute of Chemical Technology have a much more compound library (0.6 million). Also, we could use an interesting collection, which we could chose based on structure or any other categories. We chose structurally similar 2000 compounds to a well known LPA antagnonist, Ki-167427 in this study. Thus 2000 compounds similar to Ki-164727 are not small. Most compounds from Korea Research Institute of Chemical Technology are collected from commercially not available and unique compounds. We commented in line 147-149 on page 4. KA-1002 is also that case. However, we could synthesized in bulk size in our institute. We revised Line 102, “3 × 103 EBTr cells” should be “3 × 103 EBTr cells”. We also used Ki-16427, a known LPA antagonist as you recommended. (Figure 1). We revised text in line 155-156, KA-1002 was shown to decrease the induction of TNFα, IL-6, and IL-1β expression in LPA-treated CAPE cells in a dose-dependent manner. Because, we reperformed the experiment and showed dose-dependency.
